



# Three-dimensional methane distribution simulated with FLEXPART 8-CTM-1.1 constrained with observation data

Christine D. Groot Zwaaftink[1], Stephan Henne[2], Rona L. Thompson[1], Edward J. Dlugokencky[3], Toshinobu Machida[4], Jean Daniel Paris[5], Motoki Sasakawa[4], Arjo Segers[6], Colm Sweeney[3], Andreas Stohl[1]

1 Norwegian Institute for Air Research NILU, Kjeller, Norway
2 Empa, Swiss Federal Laboratories for Materials Science and Technology, Air Pollution/Environmental Technology, Dübendorf, Switzerland
3 NOAA Earth System Research Laboratory, Boulder, Colorado, USA
4 National Institute for Environmental Studies, Tsukuba, Japan
5 Laboratoire des Sciences du Climat et de l'Environnement, Gif sur Yvette, France
6 Netherlands Organisation for Applied Scientific Research (TNO), Utrecht, The Netherlands

*Correspondence to*: Christine Groot Zwaaftink (cgz@nilu.no)

**Abstract**

A Lagrangian particle dispersion model (FLEXPART CTM) is used to simulate global three-dimensional fields of trace gas abundance. These fields are constrained with surface observation data through nudging, a data assimilation method which relaxes model fields to observed values. Such fields are of interest to a variety of applications, such as inverse modelling, satellite retrievals and estimating global growth rates of greenhouse gases. Here, we apply this method to methane using 6 million model particles filling the global model domain. For each particle methane mass tendencies due to emissions based on several inventories, loss by reaction with OH, Cl, and $O(^1D)$, as well as observation data nudging were calculated. Model particles were transported by mean, turbulent and convective transport driven by 1°x1° ERA Interim meteorology. Nudging is applied at 79 surface stations, which are mostly included in the WDCGG database or JR-STATION network in Siberia. For simulations of one year (2013), we perform a sensitivity analysis to show how nudging settings affect modelled concentration fields. These are evaluated with a set of independent surface observations and with vertical profiles in North America (NOAA/ESRL) and Siberia (YAK-AEROSIB and NIES). FLEXPART CTM results are also compared to simulations from the global Eulerian model, TM5, based on optimized fluxes. Results show that nudging strongly improves modelled methane near the surface, not only at the nudging locations, but also at independent stations. Mean bias at all surface locations could be reduced from over 20 ppb to less than 5 ppb through nudging. Near the surface, FLEXPART CTM, including nudging, appears better able to capture methane molar mixing ratios than TM5 with optimized fluxes, based on a larger bias of over 13 ppb in TM5 simulations. The vertical profiles indicate that nudging affects model methane at high altitudes, yet leads to very little improvement in the model results there. Averaged from 19 aircraft profile locations in North America and Siberia, root-mean square error (RMSE) changes only from 16.3 to 15.7 ppb through nudging, while the mean absolute bias increases from 5.3 to 8.2 ppb. The performance for vertical profiles is thereby similar to TM5 simulations based on TM5 optimized fluxes where we found a bias of 5 ppb and RMSE of 15.9 ppb. With this rather simple model setup, we thus provide three-dimensional methane fields suitable for use as boundary conditions in regional inverse modelling, as prior information for satellite retrievals, and for more accurate estimation of mean mixing ratios and growth rates. The method should also be applicable to other long-lived trace gases.

## 1. Introduction

Three-dimensional (3D) global concentration fields of different trace gases, such as methane ($CH_4$), carbon dioxide ($CO_2$) or carbon monoxide (CO), are of interest for many applications. They can inform regional air quality policies, verify climate policies via a comparison of global emissions and atmospheric concentration growth rates, and are required as input for many



types of applications. For instance, 3D global concentration fields used as initial and boundary conditions for regional chemical transport models have a substantial influence on the results from these models (e.g. Tang et al., 2007; Andersson et al., 2015; Pendlebury et al., 2018). Inverse modelling of greenhouse gas emissions often requires global 3D greenhouse gas

concentrations as input (Thompson and Stohl, 2014). These and many other applications explain the popularity of 3D concentration fields produced with different tools (e.g., Carbon Tracker; Peters et al., 2007).

In-situ measurements are typically too sparse to provide global 3D concentration fields. Interpolation methods are associated with large errors in regions with few observations (e.g., in the free troposphere, over oceans, etc.). They are, therefore, rarely used for constructing 3D concentration fields for the atmosphere, while interpolation of maritime $CO_2$ measurements is more

popular (Laruelle et al., 2017). Satellite retrievals can in principle produce global coverage but generally lack sufficient temporal and/or vertical resolution, spatial coverage (especially in polar regions) and the calibration necessary to make them comparable to ground-based measurements and they can therefore contain substantial uncertainties and biases. Often, satellite observations require input of prior vertical concentration profiles in their retrieval (Schepers et al., 2012). Concentration fields can also be simulated with atmospheric chemistry transport models, yet uncertainties in the fluxes (emissions and surface

uptake) and atmospheric chemistry and transport errors in modelling will lead to mismatches between modelled and observed concentrations. In particular, if simulated concentrations are globally or regionally too high or too low compared to reality, such biases may render these concentration fields unusable. For example, for regional inverse modelling of surface emissions, a "background" concentration field biased high would lead to low (or even, for many species, unrealistic negative) emissions (Manning et al., 2003; Stohl et al, 2009). For example, an average increase by 10 ppb in the methane background leads to a

20% decrease in the Swiss national methane emissions (Henne, unpublished results).

Therefore, often a combination of (transport) models with data assimilation techniques is used to obtain more accurate model results. One prominent technique uses observation data and a data assimilation or inverse modelling approach to optimize the surface fluxes of greenhouse gases. Such methods can improve our understanding of greenhouse gas emissions and quantify sources and sinks (e.g., Meirink et al., 2008; Bergamaschi et al., 2013; Berchet et al., 2015). Simulations using the optimized

emissions will generally also improve the 3D concentration distribution compared to simulations with non-optimized emissions. The TM5 simulations used in this paper were made following this approach. However, these methods cannot correct for transport errors and errors in the chemical sinks, although such errors may partly be offset by compensating errors in optimized emissions. Furthermore, emissions are only improved for certain large regions, emission types and often at low (e.g., monthly) time resolution. Thus, to improve 3D concentration distributions, direct "correction" of simulated concentrations

using observation data is preferable over improvement of emission fluxes. A combination of both approaches can also be conceived.

Data assimilation techniques such as 4-dimensional variational assimilation (4D-var) also allow simulated concentrations to be corrected in a formal fully consistent model framework. These techniques are, however, relatively computationally expensive and may not offer substantial benefits over more simple methods. Newtonian relaxation (also known as nudging or

repeated insertion) is a much simpler data assimilation method to correct simulated concentrations with observation data (e.g., Anthes, 1974; Stauffer and Seaman, 1990). While it may lead to violations of the model's underlying physics and mass conservation, the method has gained popularity because of its computational efficiency, robustness and simple implementation. In contrast to complex data assimilation techniques such as 4D-var, which require adjoint versions of the simulation model, implementing Newtonian relaxation requires only a few lines of extra code.

Apart from the question of how observation data are used to correct model biases, there exists a separate problem with current Eulerian chemistry transport models. These models struggle with the simulation of large-scale transport of chemical species e.g., over intercontinental distances due to numerical diffusion introduced by the advection schemes (Rastigejev et al., 2010; Eastham and Jacob, 2017). In remote areas, where concentration enhancements occur mainly through transport from far-away source regions, these models are not very reliable. Lagrangian transport models, which do not suffer from numerical diffusion

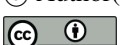



to a comparable extent, have often been used successfully to simulate pollution transport over intercontinental distances (e.g., Stohl et al., 2003).

In this study, we improve modelled concentration fields of long-lived trace gases in a Lagrangian transport model (FLEXPART CTM) by nudging simulated concentrations towards observations where and when available. We explore the method with the application to methane. Concentration fields obtained with and without nudging are evaluated based on (independent) surface

observations and vertical profiles from aircraft campaigns in different regions. Furthermore, we compare the model skill to that of a frequently used Eulerian model (TM5) using non-optimized and optimized emission fields. For this comparison we chose TM5 because this is a state-of-the-art transport model and the TM5-4DVar is the inversion framework used in Europe's Copernicus Atmospheric Monitoring Service.

A description of the transport model, nudging method and observations is given in section 2. In section 3 we will first discuss

the sensitivity of modelled concentration fields to settings of the nudging method and then evaluate model performance in comparison to surface and profile observations, and TM5 modelled concentration fields.

## 2. Methods and Data

### 2.1 FLEXPART CTM

The FLEXPART CTM model was developed at Empa based on FLEXPART (Stohl et al., 1998; 2005). FLEXPART is a

Lagrangian particle dispersion model that calculates trajectories of parcels to describe transport processes in the atmosphere. Parcels, which can represent gases or aerosols, are influenced by dry deposition and wet deposition. For the current study we used FLEXPART8-CTM1.1 (Henne et al., 2018), which is based on FLEXPART 8.0 and described in detail by Henne et al. (in preparation). Important differences between FLEXPART and FLEXPART CTM are found in the implementation of flux fields and chemistry. For the current purpose, simulations are done in domain filling mode (Stohl and James, 2004). This

means that at initialization air parcels are randomly distributed over the whole model domain (in this case global) proportionally to air density. Each parcel represents a fraction of the total atmospheric mass. In addition to an air tracer, each parcel can also carry a number of chemical species including methane. Whenever air parcels reside near the surface, methane fluxes are accounted for by changing the methane masses of the respective parcel. Methane loss through reaction with OH, Cl and O($^1$D) radicals is also accounted for in form of a pseudo first order reaction with prescribed, monthly variable concentration

fields (Henne et al., in preparation). As such, we can simulate methane concentrations if initial conditions are well represented and a spin-up time is included.

The simulations presented here are driven with methane fluxes from a combination of several inventories. For anthropogenic sources, we included fluxes from the Emissions Database for Global Atmospheric Research (EDGAR), versions v4.2 for 2000-2008, while 2008 emissions were repeated for 2009-2013. Monthly biomass burning emissions are based on the Global Fire

Emissions Database (GFED) version 3 (Van der Werf et al., 2010). Wetland emissions were obtained from the Lund-Potsdam-Jena dynamic global vegetation model with Wetland Hydrology and Methane (LPJ-WHyMe, Spahni et al., 2011). Additional sources taken into account include ocean hydrates (Houweling et al., 1999), wild ruminants (Houweling et al., 1999) and termites (Sanderson, 1996). Flux fields are averaged or interpolated to monthly intervals with 1° x 1° spatial resolution. A spin-up simulation without nudging was made over 2000-2012, by the end of which a single global scaling factor was applied

to the simulated methane molar mixing ratios derived by a comparison to surface observations, and the sensitivity analysis and evaluation was made for 2013. The first 20 days of the simulation year with nudging are also considered spin-up time and not analysed further. Meteorological input data are ERA Interim Reanalysis fields (1° x 1°, 3-hourly) from the European Centre for Medium-Range Weather Forecasts (ECMWF). Output fields are saved as daily averages at 2° x 2° spatial resolution at 24 levels. The output levels follow topography and resolution ranges from 500 m near the surface, 1000 m in the troposphere and

increasing up to 5000 m at the top level. The simulations include approximately 6 million particles each.



### 2.2 Nudging routine

In our simulations, we will test and use the nudging routine presented by Henne et al. (in preparation). For each observation, we consider a symmetrical kernel in which modelled data should be relaxed towards the observation. The weight of the kernel varies in space and time. An Epanechnikov function is used for the spatial weight of the kernel ($w_s$) for each pair of observation

($i$) and parcel ($j$):

$$w_{s,ij} = \left(1 - r_{ij}^2\right)I, \quad I = \begin{cases} 1 \text{ for } r^2 < 1 \\ 0 \text{ otherwise} \end{cases} \tag{1},$$

where, $r_{ij}$ is based on the kernel extent $h$ in directions x, y and z following:

$$r_{ij}^2 = \left(\frac{X_j - x_i}{h_{x,j}}\right)^2 + \left(\frac{Y_j - y_i}{h_{y,j}}\right)^2 + \left(\frac{Z_j - z_i}{h_{z,j}}\right)^2 \tag{2}$$

$X_j$, $Y_j$ and $Z_j$ refer to the parcel location, $x_i$, $y_i$, and $z_i$ to the observation location.

The temporal kernel has a tricubic weight function following

$$w_{t,ij} = \left(1 - \left|\frac{t_j - t_i}{h_t}\right|^3\right)^3 \tag{3}$$

where $h_t$ is the temporal kernel width, $t_j$ the current model time and $t_i$ the time of the closest valid observation. Using the spatial and temporal weight the nudging tendency is calculated as

$$\Delta m_{ji} = w_{s,ij} w_{t,ij} \frac{M_i - m_j}{\tau_i} \Delta t \tag{4}$$

where $m_j$ is the modelled mass and $M_i$ is the observed mass, which is calculated from observed mole fractions. $\tau_i$ is the nudging relaxation time scale and $\Delta t$ the model synchronisation time step. The latter should be smaller than the former to assure numerical stability. The kernel widths and nudging relaxation time scale can be set for each observation location. Different settings have been tested as shown in Table 1. In tests NF1 to NF6 we analysed the influence of specific parameters and kernel settings, which were the same at all stations. In tests NV1 to NV3 we introduced a kernel size dependent on the variability of

the observed methane concentrations over one year at each station. We thereby assumed that the kernel size should be smaller in case of larger variability (standard deviation). In all these tests, the temporal kernel width is assumed to be proportional to the spatial width. In test NW1 we reduced the temporal kernel size with increasing mean wind speed at each location, assuming air masses will reside less time around the nudging location if wind speeds are strong and observations are representative of a shorter time period. To determine the mean wind speed, we used the wind speed in 2013 at the stations interpolated from ERA

Interim data. In test NW2 we tested an influence of wind speed on the spatial extent of the kernel. Finally, in NW3 we assumed that spatial kernel size increases with wind speed, yet the temporal kernel width decreases with increased variability of the observed methane mole fractions.

### 2.3 Observations

Several data sources were used for different purposes in this study. Input data for the FLEXPART CTM simulations have been

described in section 2.1. At the surface, we included observations of $CH_4$ (reported in units of dry air mole fraction, nmol mol$^{-1}$, abbreviated ppb) from the World Data Centre for Greenhouse Gases (WDCGG, https://ds.data.jma.go.jp/gmd/wdcgg/) and EBAS (http://ebas.nilu.no/) at 85 locations and 5 locations in the JR-STATION tall tower network in Siberia (Sasakawa et al., 2010). Observations include flask-air sampled at intervals of a few days up to months and continuous measurements. All observations have been converted to the NOAA-2004 scale. NIES data were converted to NOAA-2004 scale

assuming $CH_{4\,NOAA-2004} = \left(CH_{4\,NIES} + 12.2\right)/1.0087$. Of 90 surface or tower observation locations in total, we arbitrarily



selected 11 for validation purposes that were not used for nudging. Locations are shown in Figure 1, a full list of surface stations is given in the Supplement (Table S1).

Further validation of the nudging routine was based on the comparison to aircraft measurements. These include regular vertical profiles of NOAA at 13 locations (Sweeney et al., 2015). Profiles were taken from the surface up to about 8000 m altitude in
approximately monthly intervals. Furthermore, we include methane profiles observed in Siberia during a YAK-AEROSIB campaign (Paris et al., 2008) in July 2013 and observed by NIES at monthly intervals at Surgut and Novosibirsk (Sasakawa et al., 2017) for validation. Locations are shown in Figure 1. We interpolated FLEXPART CTM data to each measurement during NOAA, NIES or YAK-AEROSIB flights. We then calculated mean profiles per month based on the vertical grid resolution of FLEXPART CTM output (24 layers).


### 2.4 TM5 simulations

To compare the performance of FLEXPART CTM with the nudging routine, we also evaluate reference and reanalysis fields of methane from the global chemistry transport model TM5 (Huijnen et al., 2010; Krol et al., 2005) with the same observations and methods. Methane inversion simulations are provided by TNO-SRON and are available through the Copernicus Atmosphere Monitoring Service (CAMS). The TM5-4DVAR inverse modelling system provides optimized methane fluxes
(e.g., Bergamaschi et al., 2013). The TM5 methane fields are averaged daily and have a 2° x 3° resolution. In our evaluation we include TM5 simulations based on a-priori information (here referred to as the TM5 reference simulation) and a simulation with optimized fluxes (referred to as TM5 reanalysis simulation) that included all NOAA surface observations. Notice that TM5 has assimilated data also from the surface stations used for evaluation and, thus, the comparison is not totally independent,
giving the TM5 reanalysis simulation a (perhaps small) advantage over the FLEXPART CTM simulations. We did not use TM5 reanalysis simulations with assimilated Greenhouse Gases Observing Satellite (GOSAT) data (Bergamaschi et al., 2009) since the bias in comparison to surface observations was larger than in the presented simulations, as shown in the Supplement (Figures S1 and S2).

### 3. Results and discussion

### 3.1 Sensitivity analysis

The nudging kernel settings such as the size, described in section 2.2, will affect the influence of observations on the methane fields. To show this influence and to find appropriate settings for our application, we performed a sensitivity analysis. The different kernel settings are listed in Table 1. Figure 2 shows the annual mean surface methane molar mixing ratios as simulated with FLEXPART CTM without nudging. Additionally, the difference in annual mean mixing ratios between the reference
simulation and two nudging simulations is shown. With relatively small nudging kernels (NF2, middle panel) some influence on global methane fields is seen. The strongest effects, however, are restricted to the nudging locations. Increasing the kernel size for background stations (NV3, bottom panel) shows that the nudging can influence simulated methane values strongly in a far larger region. The nudging appears to mostly lower modelled methane mole fractions in the Northern Hemisphere and to increase methane in the Southern Hemisphere, indicating a positive and negative bias of methane concentrations respectively.
This will later be discussed based on a comparison to observations.

A more quantitative analysis of the sensitivity to nudging kernel settings is given in Figure 3. We compare observed and modelled daily averaged methane concentrations in 2013 through the coefficient of determination ($r^2$), root mean squared error (RMSE) and bias for nudged FLEXPART CTM simulations and the reference simulation at all surface locations (including both stations used for the nudging and independent validation stations). Generally, model performance in terms of root mean
squared error, correlation and bias improves for all tested nudging kernel settings compared to the reference simulation.



Already for a rather small kernel size (NF1), the coefficient of determination increases, while there remains a relatively large spread in model bias. It also appears that the chosen relaxation time (NF4 vs NF3 or NV3 vs NV2) affects the coefficient of determination. Especially large changes are seen for the temporal kernel width (NW3 vs NW2 and NF5 vs NF3).

The representativeness of a station is directly linked to the variability of the observed methane concentrations. If the variability

is high, this may indicate that the station is often influenced by emissions in its vicinity or by small-scale transport phenomena. Such a station should exert less weight on the simulated concentration field than a station with greater representativeness and its area (and/or time period) of influence should be smaller. Therefore, in another set of tests (NV1-3 and NW3) we assumed that the variability of observed concentrations (expressed in terms of the annual standard deviation) limits either the temporal or spatial kernel width. Unlike tests NF1-NF6, the kernel size is thereby no longer identical for all locations. Based on the

small values of RMSE it appears that changing the spatial kernel size based on the observed variability can improve the performance of the nudging routine (NV1 and NV3). The mean RMSE in the reference case is 35.3 ppb for all stations and 23.6 ppb for validation stations, and is reduced to 10.3 ppb for all and 18.7 ppb for validation stations in the NV3 simulation. The correlation improves mostly at nudging stations, as can be concluded from an increase of mean $r^2$ for all stations from 0.54 in the reference to 0.92 in the NV3 simulation, yet at validation stations remains at 0.62 for REF and NV3.

We are particularly interested if the nudging routine can remove some of the bias of modelled methane simulations, as bias is detrimental for many applications (e.g., inverse modelling). In Figure 4 we therefore show the model bias and coefficient of determination at surface stations for a selection of sensitivity simulations. In the reference simulation, methane concentrations tend to be underestimated in the Southern Hemisphere and overestimated in the Northern Hemisphere by FLEXPART CTM with the used emission inventories. The nudging routine removes most of the model bias at many stations. We found a large

variation of observed methane in Siberia, at the stations of the JR-STATION network. At some of these stations daily mean mixing ratios reach values exceeding 2000 ppb (Sasakawa et al., 2010). The stations are located in taiga, steppe and wetland biomes. Sasakawa et al. (2010) showed that during summer methane emissions from wetlands and during winter from fossil fuel extraction can explain most of the variation in methane values in this region. Also biomass burning is locally contributing to elevated methane. It appears that in this region, the model benefits from nudging kernel sizes related to the standard deviation

of observations (compare NV3 to NF3). For simulation NV3, the nudging kernel sizes are relatively small and there is less overlap of the different kernels in this region, allowing for a larger spatial variability in simulated concentrations.

Bias and $r^2$ shown in Figure 4 summarize model performance at many stations. To get a better understanding of how nudging changes the modelled methane concentrations we discuss the annual cycle at two surface stations (Figure 5 and 6). As an example of a nudging location we look at the remotely located Palmer Station, Antarctica (PSA, -64.00°E, -64.92°N). Here,

observations include approximately weekly flask-air samples. Modelled and observed mixing ratios in 2013 are shown as time series and a scatter plot (Figure 5). Additionally, model performance statistics for PSA are given in the table in Figure 5. The figures and table show that correlation between model and observations is already large for the reference simulation as the seasonal cycle is well captured by the model, but the bias can be reduced through nudging. For the smallest nudging kernel size (NF1), the improvement is limited, from -8.5 ppb (REF) to -8.4 ppb (NF1). For all other nudged simulations, however,

the bias is reduced to less than -2.3 ppb. The time series reveal more differences. Since the variation of observed methane mixing ratios at PSA is relatively small, the nudging kernel in simulation NV3 was large (over 10 degrees radius). In case of small nudging kernels (NF3), methane deviates from the reference simulation only during measurement periods but nearly returns to the reference values in between measurements. For larger kernels (NV3), the methane concentrations are kept at a level similar to the observations also in between observations. Given the remote site location and the lack of local emissions

this scenario appears more realistic. This behaviour is particular to remote stations and rather extreme at PSA, where the reference model has a pronounced bias and there are no other observations to correct it. At other stations, the influence of other close-by nudging stations is generally larger and better results are obtained for small kernels as well. Besides the kernel settings



the number of nudging stations included in a simulation will thus also affect model performance, as will be discussed at the end of this section.

Another station for which we show observations and simulations in more detail is Heimaey (ICE, Figure 6). Heimaey is located close to the southern coast of Iceland and is an independent station where no nudging is applied in FLEXPART CTM simulations. At this location, the reference simulation has a relatively large bias of over 28 ppb. With the introduction of nudging at other stations, the bias is reduced to approximately 6 ppb and also all other statistical parameters improve (see table in Figure 6). Most improvement, both in correlation and bias reduction, was seen in simulation NV3. In this simulation the

relatively large kernel given to background stations helps to reduce bias globally. In the time series (Figure 6, left) it can be seen that model performance at the start of the year is worse than at the end of the year. Although the deviation between reference and nudged simulations occurs from the start, the difference between several simulations accumulates during the simulation period as more observations are included. We show the simulations over the complete nudging period to demonstrate this, but did not include the first 20 days with nudging in the model performance calculations. Deviations can of

course also increase if the model performs worse in particular periods, for instance due to errors in emissions, inducing stronger nudging effects.

To demonstrate that the nudging routine generally improves model results throughout the domain, we evaluate the bias at all independent validation stations in Table 2. Effects are smaller than at many nudging stations shown in Figure 4. Nonetheless, there is a clear improvement at some stations, even though they are remote, such as USH (Ushuaia, Tierra del Fuego,

Argentina). Table 2 also shows that FLEXPART CTM model performance at some stations can decrease depending on nudging kernel settings. This is the case at THD (Trinidad Head, California, United States of America) and YAK1 (Yakutsk, Siberia, Russia). While methane mixing ratios were slightly underestimated at THD in the reference simulation, an overestimation was observed at close-by nudging locations. In the nudging simulations methane concentrations were thus decreased in this region and underestimation at the independent station increased. At Yakutsk measurements are made at a low inlet (11 m, here called

YAK1) and a high inlet (77 m, YAK2). Nudging appears to decrease bias of the methane values at the high inlet, but not at lower altitude. A large bias both in reference and nudged simulations is seen at BKT (Bukit Koto Tabang, Indonesia). This station is influenced by near-by biomass burning emissions and a sea-breeze system that is difficult to resolve in the FLEXPART simulations (Henne et al., in preparation).

JFJ is a high-altitude station and the strongly decreased bias of modelled mixing ratios indicates that surface nudging has a

beneficial influence also at higher altitudes (observations from JFJ were not used in nudging). To verify the influence of kernel settings on methane mixing ratios at high altitude we selected two locations with NOAA/ESRL vertical profiles.

Nudging at the surface changes modelled methane concentrations up to heights of 8 km (Figure 7). At ACG (Alaska Coast Guard, Alaska, United States of America) differences due to surface nudging even appear largest at heights between 2 and 6 km. The number of nudging observations is limited in this region and differences are, therefore, mostly due to nudging in well

observed regions further south, followed by northward transport accompanied by isentropic uplift of those air masses to higher altitudes. Similarly, Sweeney et al. (2015) showed based on vertical profiles of $CO_2$ that air transported to this region, from the mid-Pacific, is well mixed throughout the observed column (<8000 m). In a region with multiple nudging locations such as DND (Dahlen, North Dakota, United States of America) on the other hand, mainly the methane concentrations near the surface are affected by nudging. As for surface validation stations, improvements are seen mostly in RMSE and bias rather

than correlation (Table 3).

Finally, we noted earlier that, besides nudging kernel properties, the number of nudging observations included in the simulation can affect the model performance. We, therefore, tested model performance for simulations with different numbers of particles and where some of the nudging locations in NV3 were removed. As expected, reducing the number of particles somewhat increases the noise in the simulation but does not strongly influence overall model performance (not shown), because as long

as the parcels in the simulation are well-mixed the same fraction of particles will be influenced by nudging. We expect a larger



difference in the case where we randomly remove nudging stations from our simulations. We tested this for the kernel settings used in NV3, with 50% and 80% of the nudging stations included. A summary of model performance results is given in Table 4. Looking at all stations, correlation increases and RMSE decreases with an increase in number of stations. As already discussed, the influence on correlation between model and observations at independent sites is limited. RMSE at independent
validation sites does decrease if 50% or 80% of the nudging locations are included, but the final 20% do not improve results here. With 80% of the nudging locations the background methane mixing ratios were thus already improved globally, or the nudging stations most relevant for the validation stations were already included in the 80%.

### 3.2 Model performance

Based on results shown in previous sections we selected the NV3 run to provide the best nudged FLEXPART CTM methane distribution. For further analysis of model performance we will only use this simulation. We will evaluate this best case simulation in comparison with the methane field simulated by the Eulerian TM5 model (see section 2.4).

### *Surface mixing ratios*

As for FLEXPART-CTM in Figure 4, we show bias and correlation values for one year of TM5 simulations at all surface stations (Figure 8). We include TM5 simulations that are based on a-priori information (TM5 REF) and TM5 reanalysis simulations that include fluxes optimized by the TM5-4DVAR system (e.g. Bergamaschi et al., 2010). When comparing the reference versions of FLEXPART CTM and TM5, similar biases can be found for both models, although FLEXPART CTM's performance with mean RMSE equal to 35.3 ppb and mean absolute bias of 20.9 ppb appears slightly poorer than that of TM5
with mean RMSE equal to 33.9 ppb and mean absolute bias of 17.3 ppb (also see Table 5). On the other hand, correlation is higher for FLEXPART CTM ($r^2$= 0.54) than for TM5 ($r^2$= 0.39). Differences may be due to the use of different emission inventories but can also be related to other modelling aspects (e.g., differences in methane lifetime). More interesting however, are the performance of FLEXPART CTM when the nudging routine is switched on and TM5 based on optimized fluxes, shown in Figure 8 and Table 5. Especially if nudging stations and validation stations are both considered, the performance of the
nudged FLEXPART CTM simulation (NV3) is better than that of the TM5 simulations. Mean bias of all stations was reduced to 4.9 ppb in NV3, but 13.3 ppb in the optimized TM5 simulation. At independent stations (for FLEXPART CTM), methane mixing ratios are on average biased by 8.8 ppb in NV3 and 8.1 ppb in TM5 reanalysis. It should be noted that most of the nudging data are also used in the reanalysis version of TM5. In fact, data from independent NOAA surface stations (diamonds in Figure 8) such as ICE, USH, AZR and GMI are not used in FLEXPART CTM, but are ingested in the TM5 reanalysis
system and better performance by TM5 may, therefore, be expected. Observations from the Yakutsk tower are not used in either of the simulations. Here, the model bias is lower in the nudged FLEXPART CTM simulations. This is probably also related to the inclusion of data from the JR-STATION network in FLEXPART CTM, which are not included in TM5. It appears that both models, with optimization, are well able to capture background concentrations in the Southern Hemisphere. In the northern mid-latitudes, deviations from observations are generally largest. This is also the region with the most
observations and spatial variability appears large. Here, the models are, with the current setup, not able to capture this large spatial variability and observations are representative of small regions only. This is further demonstrated in a comparison of averaged zonal mean values from FLEXPART CTM and TM5 simulations and zonal mean values from the GLOBALVIEW-CH4 product in Figure 9. GLOBALVIEW-CH4 extends surface measurements of methane in time and space to provide model independent trace gas climatology (Masarie and Tans, 1995; GLOBALVIEW-CH4, 2009). Differences between the three
products are largest from the equator to the northern mid-latitudes. The GLOBALVIEW-CH4 product is probably more representative of a maritime baseline and biased low in the Northern Hemisphere, partly explaining the larger difference here between both models and GLOBALVIEW-CH4.



*Aircraft profiles*

To assess model performance above the surface we use vertical profiles obtained from aircraft measurements at 16 locations. 12 of these are located in North America and are part of the NOAA/ESRL aircraft program, one is also part of NOAA but is in the region of the Cook Islands in the Central-Southern Pacific Ocean (Rarotonga). In Siberia, we obtained data from the YAK-AEROSIB campaign and regular NIES profiles at Surgut and Novosibirsk. We show mean profiles in 2013 at each location in Figure 10. Besides comparing FLEXPART and observations, we again also include results of TM5 reference and

reanalysis (or 4DVAR) output. As was also shown before based on vertical profiles at two stations (Figure 7), nudging FLEXPART CTM simulations at the surface influences vertical profiles throughout the troposphere. In North America, many surface observations used for nudging FLEXPART CTM are also included in the TM5 Reanalysis simulations. Good results are therefore obtained with both models at some locations, such as DND (Dahlen) or CMA (Cape May). It appears that FLEXPART CTM tends to underestimate methane concentrations at altitudes above 6000 m (see for instance SCA, WBI and

YAK-AEROSIB). This may indicate that the exchange between troposphere and stratosphere is too strong in FLEXPART CTM, or that the stratospheric methane may be underestimated. Obviously, the performance of both models varies with altitude, as will be discussed later. An especially large difference between FLEXPART CTM and TM5 is seen at Novosibirsk, even though this does not appear related only to the inclusion of Siberian data in the FLEXPART CTM nudging routine. This may be due to the use of different emission inventories.

The profiles shown in Figure 10 are averages of all available monthly profiles per site. Model performance of FLEXPART CTM and TM5 varies throughout the year however, and as an example we show seasonally averaged profiles at ESP (Estevan Point, British Columbia, Canada) in Figure 11. Both models appear to benefit from the nudging and reanalysis techniques, respectively, with improved concentrations, especially near the surface, compared to reference simulations. For both models, strong deviations from measurements occur in autumn with very similar profiles for TM5 and nudged FLEXPART CTM. The

nudging appears to improve modelled FLEXPART CTM concentrations up to roughly 3 km altitude in all seasons except for autumn.

  To assess if modelled vertical profiles deviate systematically from observed vertical profiles, we split the profiles into three altitude ranges, below 2000 m altitude, from 2000 up to 6000 m altitude, and above 6000 m. In Figure 12, the bias of each model is shown for all NOAA profiles in North America for each of the altitude ranges. As was already indicated by the

average profiles in Figure 10, FLEXPART CTM tends to underestimate methane at altitudes above 6000 m. The nudging routine near the surface partly enhances this bias in this altitude range. In TM5, values are rather overestimated at this altitude and an improvement between the reference and reanalysis run is visible. At the altitude range between 2000 and 6000 m, FLEXPART CTM performs better than at upper levels. For TM5 there seems to be a small overestimation of methane concentrations at this altitude range. Near the surface, below 2000 m, a clear impact of the nudging routine for FLEXPART

CTM is shown. For TM5, strong positive as well as negative biases occur near the surface and are larger in the reanalysis simulation. For this region, it appears that the FLEXPART CTM nudging routine improves near-surface concentration fields, while the TM5 reanalysis scheme is better able to improve high-altitude concentrations. Similarly, Bergamaschi et al. (2013) showed for a comparison between TM5 modelled methane concentrations and NOAA aircraft measurements during 2003 to 2010, that RMSE in the boundary layer was larger than in the free troposphere.

The underestimation of methane concentrations at high altitude by FLEXPART CTM means that the bias of average profiles is mostly negative (Table 6), and averaged absolute bias of all profiles increases in case of nudging. The removal of methane near the surface can cause too low values at higher altitudes. Looking at RMSE, both models show similar values (~16 ppb) for average profiles (Table 7). Although nudging was particularly useful to reduce the bias near the surface, as we concluded based on surface observations, it is less effective at improving vertical profiles above about 2000 m. Nonetheless, it appears

that FLEXPART CTM nudged concentration profiles are of similar accuracy as alternatives available at present.





**Conclusions**

We presented a nudging routine in combination with FLEXPART CTM to provide three-dimensional fields of long-lived trace gas abundance constrained with surface observations. In particular, this study focuses on the optimization of methane. However, with the right settings this frame work could be used for other species.

In a sensitivity analysis, we showed that modelled methane near the surface greatly improves through the inclusion of observations at 79 locations. To show that the observations improve the concentration fields not only locally, but globally, we compare model results to independent observations. These were at the surface (11 sites) as well as vertical profiles from aircraft campaigns (19 sites). For very small kernels (0.5° width), nudging was too weak and not much improvement was seen. Large kernels are suitable for background stations, but for stations with large variability in the observations a smaller kernel leads to

better results. The number of observations included in the simulation also influences model performance. We concluded that nudging kernel settings of test NV3 (see Table 1) were best to improve modelled methane concentrations. This setting is based on a kernel whose bandwidth is dependent on the observed methane variability over one year at a site.

A comparison with methane fields from TM5 simulations was made. We included reference TM5 simulations, based on bottom-up emission information, as well as reanalysis simulations that use optimized emission fluxes (TM5-4DVAR). Results

showed that in the upper troposphere FLEXPART CTM underestimates methane and best results are obtained near the surface. In contrast, TM5 appears to be better in the upper atmosphere, yet model performance near the surface is poorer. The mean RMSE of all averaged vertical profiles was 15.7 ppb for FLEXPART CTM (nudged), yet 15.9 ppb for TM5 (reanalysis). For all surface stations mean RMSE values were 10.3 ppb and 29.7 ppb for FLEXPART CTM and TM5 respectively and at independent surface stations values were 18.7 ppb and 20.6 ppb, respectively. It should be noted, however, that the independent

stations are only strictly independent for FLEXPART CTM simulations but are partly included in TM5 assimilation runs. In future experiments it might be useful to use the same data selections for the FLEXPART nudging and the TM5 4D-var inversion. It could also be interesting to drive the FLEXPART nudging with a posteriori emissions from the TM5 inversion, to see the differences in how the models simulate 3D concentrations. Increases in correlation between observations and modelled values at surface stations are considerable at nudging locations, but low at independent validation stations.

In summary, we can simulate three-dimensional fields of methane with a Lagrangian model constrained with observations through nudging. We have shown that in the troposphere this simple assimilation technique, which is computationally inexpensive, can give similarly good results as a Eulerian chemistry transport model combined with a computationally expensive 4-dimensional variational data assimilation scheme. Near the surface, the Lagrangian model with nudging even seems to outperform the Eulerian model with data assimilation.

**Code and data availability**

The FLEXPART 8-CTM1.1 source code is available under https://doi.org/10.5281/zenodo.1249190 (Henne et al., 2018). Modelled methane concentration fields from the nudged NV3 FLEXPART CTM simulation are currently available for 2013 (https://folk.nilu.no/~christine/ch4.html) and updates will be added. Other model results discussed here are available upon request.

**Acknowledgements**

A large set of observations was used in our simulations. We thank all those involved in the EBAS and WDCGG efforts, and those who have contributed by operating sites, performing chemical analysis and making the data publically available in the data bases.





We acknowledge the following people, institutes or projects for providing observations: AGAGE, Chris M. Harth and Prof.
Ray Weiss at Scripps Institution of Oceanography (SIO), Prof. Ron Prinn at Massachusetts Institute of Technology (MIT),
Paul B. Krummel at CSIRO Oceans & Atmosphere, Dr. Ray H. Wang at Georgia Institute of Technology, Prof. Simon
O'Doherty and Dr. Kieran Stanley at the University of Bristol. AGAGE operations for the methane data used here were
primarily supported by NASA grants to MIT (NNX11AF17G) and SIO (NNX11AF15G, NNX11AF16G). The operation of
all UK DECC Network stations, TAC, RGL and TTA was funded by the UK Department of Business, Energy and Industrial
Strategy (contract TRN1028/06/2015) with additional funding at Mace Head, Ireland and Ragged Point, Barbados under
NASA contract NNX16AC98G through MIT with a sub award 5710004056 to University of Bristol, and NOAA contract RA-
133R-15-CN-0008 for Ragged Point; Cathrine Lund Myhre (NILU); China Meteorological Administration; Climate Science
Centre – CSIRO Oceans & Atmosphere; Agency for Meteorology, Climatology and Geophysics (BMKG); Direccion
Meteorologica de Chile; EMEP; Environment and Climate Change Canada; Federal Environment Agency, Austria; Federal
Environment Agency, Germany; Institute of Arctic and Alpine Research at the University of Colorado, Boulder, funded by
US National Science Foundation grant AON 1108391; Italian National Agency for New Technology, Energy and the
Environment; Izana Atmospheric Research Center; Meteorological State Agency of Spain; Japan Meteorological Agency,
Korea Meteorological Administration; National Institute of Water & Atmospheric Research Ltd. (NIWA), New Zealand;
Ricerca sul Sistema Energetico – RSE S.p.A.; South African Weather Service; Swiss Federal Laboratories for Materials
Science and Technology (Empa); University of Malta and University of Urbino.
This study was funded by the Norwegian Research Council as part of ICOS-Norway (project 245927) and was part of the
Nordic Centre of Excellence eSTICC (eScience Tools for Investigating Climate Change in Northern High Latitudes) funded
by Nordforsk (grant 57001).

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





**Table 1 Overview of kernel settings in the sensitivity analysis. The spatial width in x-direction is equal to the width in y-direction (in meters).** $\sigma_{obs}$ **is the standard deviation of observations in two years at each nudging location,** $\sigma_{max}$ **is the maximum value of** $\sigma_{obs}$ **from all nudging locations, h$_{min}$ is 1° and** $\bar{u}$ **is the mean wind speed at the nudging location based on ERA Interim data.**

| Sim | Spatial width ($h_y$, °) | Height ($h_z$, m) | Temporal width ($h_t$, s) | Relaxation time ($\tau_n$, s) |
|---|---|---|---|---|
| Ref | - | - | - | - |
| NF1 | 0.5 | 250 | $3 \cdot 3600 \cdot h_y/h_{min}$ | 3600 |
| NF2 | 2° | 500 | $86400 h_y/h_{min}$ | 7200 |
| NF3 | 2° | 500 | $86400 h_y/h_{min}$ | 3600 |
| NF4 | 2° | 250 | $86400\ h_y/h_{min}$ | 7200 |
| NF5 | 2° | 500 | $43200\ h_y/h_{min}$ | 7200 |
| NF6 | 4° | 500 | $86400\ h_y/h_{min}$ | 7200 |
| NV1 | $\sigma_{max}/\sigma_{obs} \cdot h_{min}$ (~1-20 °) | 500 | $86400\ h_y/h_{min}$ | 7200 |
| NV2 | $\sigma_{max}/\sigma_{obs} \cdot h_{min}$ (~1-20 °) | 300 | $86400\ h_y/h_{min}$ | 7200 |
| NV3 | $\sigma_{max}/\sigma_{obs} \cdot h_{min}$ (~1-20 °) | 300 | $86400\ h_y/h_{min}$ | 3600 |
| NW1 | $\sigma_{max}/\sigma_{obs} \cdot h_{min}$ (~1-20 °) | 300 | $111 \cdot 10^3 \cdot h_y /\bar{u}$ (~6 hours - 4 days) | 3600 |
| NW2 | $h_t \bar{u}/111 \cdot 10^3$ (~2-9°) | 300 | 24*3600 | 3600 |
| NW3 | $86400 \cdot \bar{u}/111 \cdot 10^3$ (~2-9°) | 300 | $24 \cdot 3600 \sqrt{\overline{\sigma_{obs}}}/\sigma_{obs}$ | 3600 |




**Table 2 Bias of modelled methane (ppb) at independent surface validation stations. The mean value is the mean of absolute bias value at all stations listed. Settings of FLEXPART CTM simulations are given in Table 1. Station locations are shown in Figure 1.**

|      | REF   | NF1   | NF2   | NF3   | NF4   | NF5   | NF6   | NV1   | NV2   | NV3   | NW1   | NW2   | NW3   |
|------|-------|-------|-------|-------|-------|-------|-------|-------|-------|-------|-------|-------|-------|
| YAK1 | 5.6   | 4.3   | 1.3   | 1.1   | 1.6   | 2.8   | -1.1  | -11.2 | -10.1 | -11.4 | -8.2  | 0.3   | 1.4   |
| YAK2 | 12.7  | 11.3  | 8.4   | 8.3   | 8.6   | 9.7   | 5.9   | -4.1  | -3.1  | -4.3  | -1.1  | 7.4   | 8.5   |
| USH  | -9.1  | -9.0  | -7.3  | -6.9  | -8.1  | -7.9  | -3.7  | -0.8  | -1.0  | -0.6  | -1.4  | -3.9  | -4.4  |
| BHD  | -7.5  | -7.9  | -4.0  | -3.0  | -5.6  | -5.6  | 1.7   | 6.1   | 4.7   | 6.1   | 5.0   | -0.9  | -2.3  |
| BKT  | -38.5 | -38.7 | -39.3 | -39.5 | -39.6 | -38.9 | -39.5 | -39.4 | -38.9 | -39.3 | -37.8 | -40.1 | -39.3 |
| GMI  | -5.2  | -5.6  | -6.4  | -6.3  | -6.6  | -6.5  | -5.9  | -4.9  | -6.0  | -5.8  | -4.1  | -6.5  | -6.0  |
| KUM  | -2.5  | -1.6  | 1.6   | 2.7   | -1.2  | 1.4   | 3.2   | 4.6   | 2.2   | 3.6   | 4.8   | 3.1   | 3.2   |
| AZR  | 12.4  | 11.8  | 8.0   | 6.8   | 8.4   | 7.9   | 5.6   | -3.4  | -2.5  | -3.3  | -1.2  | 3.9   | 3.8   |
| THD  | -0.8  | -1.5  | -4.7  | -5.1  | -3.9  | -4.4  | -7.6  | -11.9 | -12.0 | -12.0 | -10.7 | -9.0  | -8.9  |
| CBA  | 18.4  | 18.3  | 12.5  | 12.2  | 13.6  | 14.4  | 9.6   | -1.0  | -1.0  | -1.7  | 1.9   | 11.0  | 11.2  |
| ICE  | 28.4  | 27.2  | 21.0  | 20.1  | 22.0  | 22.1  | 16.5  | 6.5   | 7.2   | 6.2   | 7.1   | 16.8  | 17.4  |
| JFJ  | 24.2  | 21.9  | 14.5  | 14.8  | 15.5  | 14.5  | 14.7  | 10.8  | 10.6  | 10.7  | 9.8   | 13.2  | 11.7  |
| Mean | 13.8  | 13.2  | 10.8  | 10.6  | 11.2  | 11.3  | 9.6   | 8.7   | 8.3   | 8.8   | 7.8   | 9.7   | 9.8   |

**Table 3 Model performance at ACG (Alaska Coast Guard, USA) and Dahlen (DND) in terms of $r^2$, RMSE and bias (ppb) of a selection of simulations according to table 1.**

|      | ACG   |       |       | DND   |       |       |
|------|-------|-------|-------|-------|-------|-------|
|      | $r^2$ | RMSE  | bias  | $r^2$ | RMSE  | bias  |
| REF  | 0.50  | 14.89 | 5.36  | 0.53  | 24.17 | 7.55  |
| NF1  | 0.60  | 13.17 | 4.42  | 0.55  | 23.13 | 6.91  |
| NF3  | 0.59  | 12.38 | 2.35  | 0.56  | 19.71 | 2.69  |
| NV3  | 0.58  | 13.73 | -6.46 | 0.58  | 18.49 | -5.54 |
| NW3  | 0.58  | 13.73 | 2.21  | 0.57  | 18.73 | 0.70  |

**Table 4 Average of coefficient of determination, root-mean-square error and absolute bias at all surface stations and at surface validation stations only for different FLEXPART CTM simulations. The percentage refers to the number of nudging locations that were included in the simulation.**

|         | $R^2$ |            | RMSE |            | Bias |            |
|---------|-------|------------|------|------------|------|------------|
|         | all   | validation | all  | validation | all  | validation |
| REF     | 0.54  | 0.62       | 35.3 | 23.6       | 20.9 | 13.8       |
| NV3 50% | 0.74  | 0.63       | 18.6 | 19.5       | 9.6  | 9.0        |
| NV3 80% | 0.85  | 0.62       | 13.9 | 18.7       | 6.9  | 8.3        |
| NV3     | 0.92  | 0.62       | 10.3 | 18.7       | 4.9  | 8.8        |



**Table 5 Mean values of r², RMSE and absolute bias for different FLEXPART CTM and TM5 simulations at surface stations for all stations or independent validation stations only.**

| | R2 | | RMSE | | Bias | |
|---|---|---|---|---|---|---|
| | All | Validation | All | Validation | All | Validation |
| REF | 0.54 | 0.62 | 35.3 | 23.6 | 20.9 | 13.8 |
| NF1 | 0.62 | 0.62 | 30.1 | 23.3 | 17.8 | 13.2 |
| NF3 | 0.92 | 0.63 | 9.4 | 20.6 | 4.9 | 10.6 |
| NV3 | 0.92 | 0.62 | 10.3 | 18.7 | 4.9 | 8.8 |
| NW3 | 0.78 | 0.59 | 18.4 | 20.2 | 9.6 | 9.8 |
| TM5 REF | 0.39 | 0.41 | 33.9 | 22.2 | 17.3 | 7.9 |
| TM5 RA | 0.46 | 0.54 | 29.7 | 20.6 | 13.3 | 8.1 |

**Table 6 Bias (ppb) of averaged methane profiles at NOAA and NIES profiles sites and during the YAK-Aerosib flight campaign. Bias is determined with equal weight of each vertical layer.**

| | FLEXPART-CTM REF | FLEXPART-CTM NV3 | TM5 REF | TM5 RA |
|---|---|---|---|---|
| ACG | 5.4 | -6.5 | 13.6 | -2.5 |
| CAR | -3.6 | -8.8 | 8.7 | 7.0 |
| CMA | 0.6 | -8.2 | 0.7 | 0.5 |
| CRV | 4.1 | -9.7 | 2.6 | -8.8 |
| DND | 7.5 | -5.5 | 2.2 | -5.1 |
| ESP | 6.0 | -8.2 | 4.2 | -5.4 |
| ETL | 14.2 | -2.4 | 6.2 | -0.3 |
| HIL | -5.7 | -15.7 | -2.9 | -2.8 |
| LEF | 5.6 | -1.9 | 0.2 | -0.6 |
| NHA | -0.8 | -11.0 | 2.4 | 1.0 |
| PFA | 2.3 | -10.2 | 1.4 | -6.8 |
| RTA | -11.0 | -3.5 | -3.8 | 0.9 |
| SCA | -8.1 | -14.2 | 3.0 | 2.4 |
| TGC | -3.8 | -6.9 | 11.2 | 9.0 |
| THD | -2.9 | -11.6 | 9.3 | -0.1 |
| WBI | 2.4 | -9.1 | 5.2 | 6.0 |
| YAK-Aerosib | 3.6 | -13.6 | 3.0 | -5.4 |
| Novosibirsk | -3.4 | -8.3 | 18.3 | 19.5 |
| Surgut | 10.2 | 0.3 | -2.6 | -10.9 |
| *Mean of absolute values* | *5.3* | *8.2* | *5.3* | *5.0* |





**Table 7 RMSE (ppb) of averaged methane profiles at NOAA and NIES profiles sites and during the YAK-Aerosib flight campaign. Bias is determined with equal weight of each vertical layer.**

|  | FLEXPART-CTM REF | FLEXPART-CTM NV3 | TM5 REF | TM5 RA |
|---|---|---|---|---|
| ACG | 14.9 | 13.7 | 19.6 | 10.4 |
| CAR | 11.6 | 14.5 | 20.6 | 15.4 |
| CMA | 15.4 | 13.6 | 17.8 | 15.1 |
| CRV | 19.3 | 20.4 | 24.8 | 20.2 |
| DND | 24.2 | 18.5 | 22.0 | 19.1 |
| ESP | 19.6 | 14.1 | 17.7 | 14.6 |
| ETL | 23.1 | 13.2 | 20.2 | 19.9 |
| HIL | 19.3 | 21.4 | 24.9 | 25.2 |
| LEF | 11.9 | 11.8 | 19.6 | 12.1 |
| NHA | 14.8 | 17.1 | 17.6 | 15.5 |
| PFA | 15.7 | 16.6 | 15.1 | 12.8 |
| RTA | 13.3 | 7.7 | 7.8 | 6.8 |
| SCA | 18.9 | 21.6 | 21.8 | 18.6 |
| TGC | 13.0 | 14.7 | 17.9 | 17.4 |
| THD | 15.5 | 17.8 | 20.3 | 15.5 |
| WBI | 14.9 | 13.8 | 10.7 | 12.8 |
| YAK-Aerosib | 18.6 | 20.4 | 13.2 | 13.8 |
| Novosibirsk | 6.1 | 10.5 | 22.4 | 23.8 |
| Surgut | 19.0 | 17.5 | 5.8 | 12.3 |
| *Mean* | *16.3* | *15.7* | *17.9* | *15.9* |




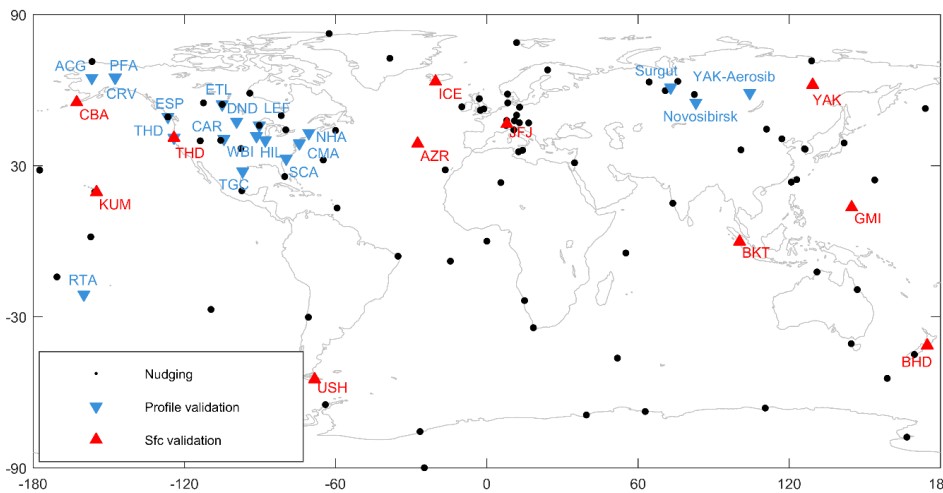

**Figure 1 Map of surface and profile locations used for nudging or validation. For profiles that span a larger domain only the centre location is indicated.**





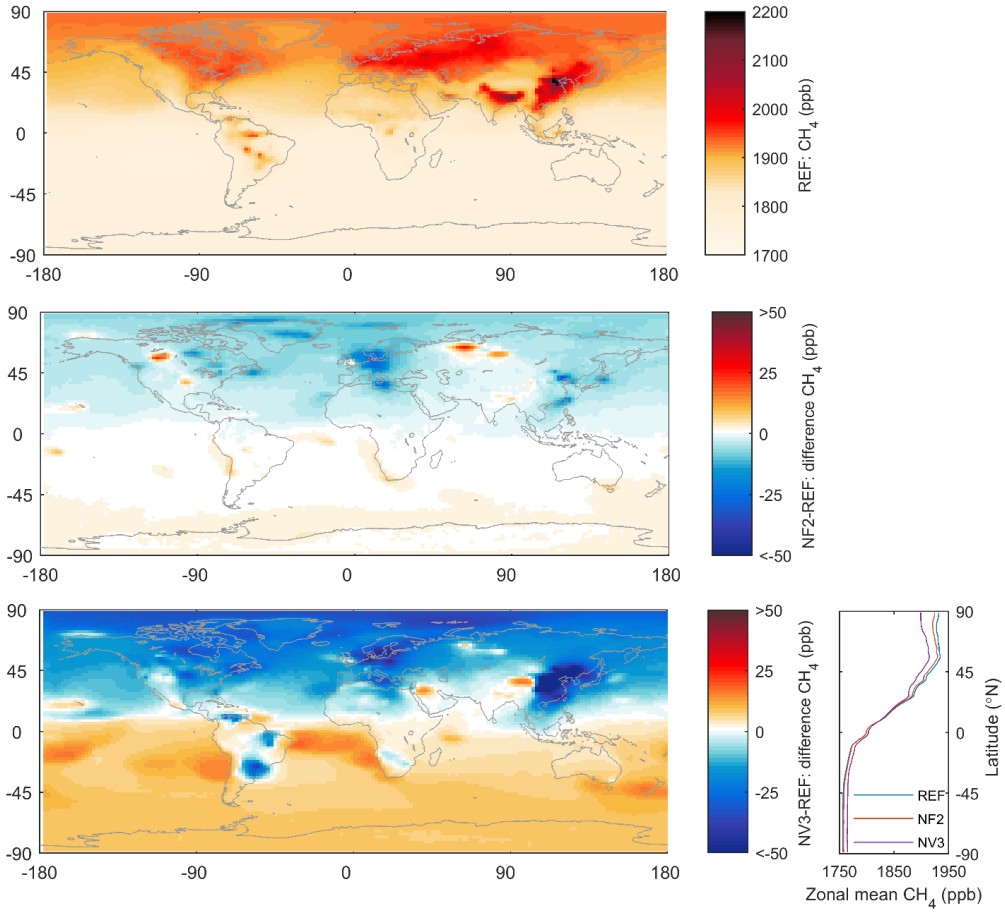


**Figure 2 Simulated annual mean surface methane (ppb) for 2013 in the FLEXPART CTM reference simulation (top) and the difference in annual mean surface methane for FLEXPART CTM NF2 (middle) and NV3 (bottom, left) compared to the reference simulation. Additionally, zonal averages of surface methane (ppb) for all three simulations are also shown (bottom, right).**




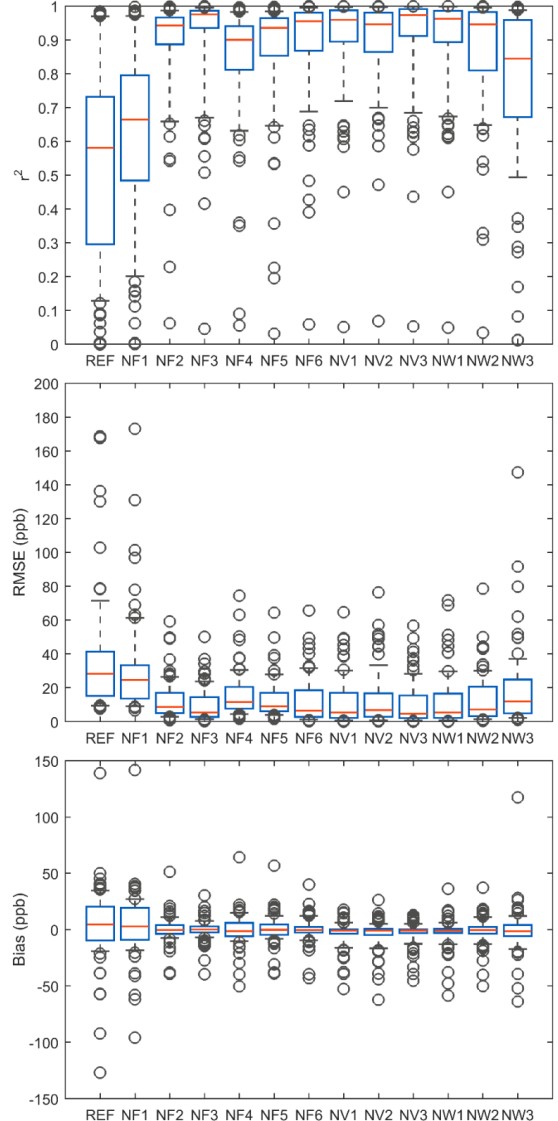

**Figure 3 FLEXPART CTM model performance based on boxplots of coefficient of determination, RMSE and bias in daily average methane model fractions at surface stations for different kernel settings (see Table 1). Independent validation stations are included. The red line in the boxplots indicates the median, the bottom and top box edges are the 25th and 75th percentiles, and whisker ends are at 9th and 91st percentiles. The circles are outliers.**




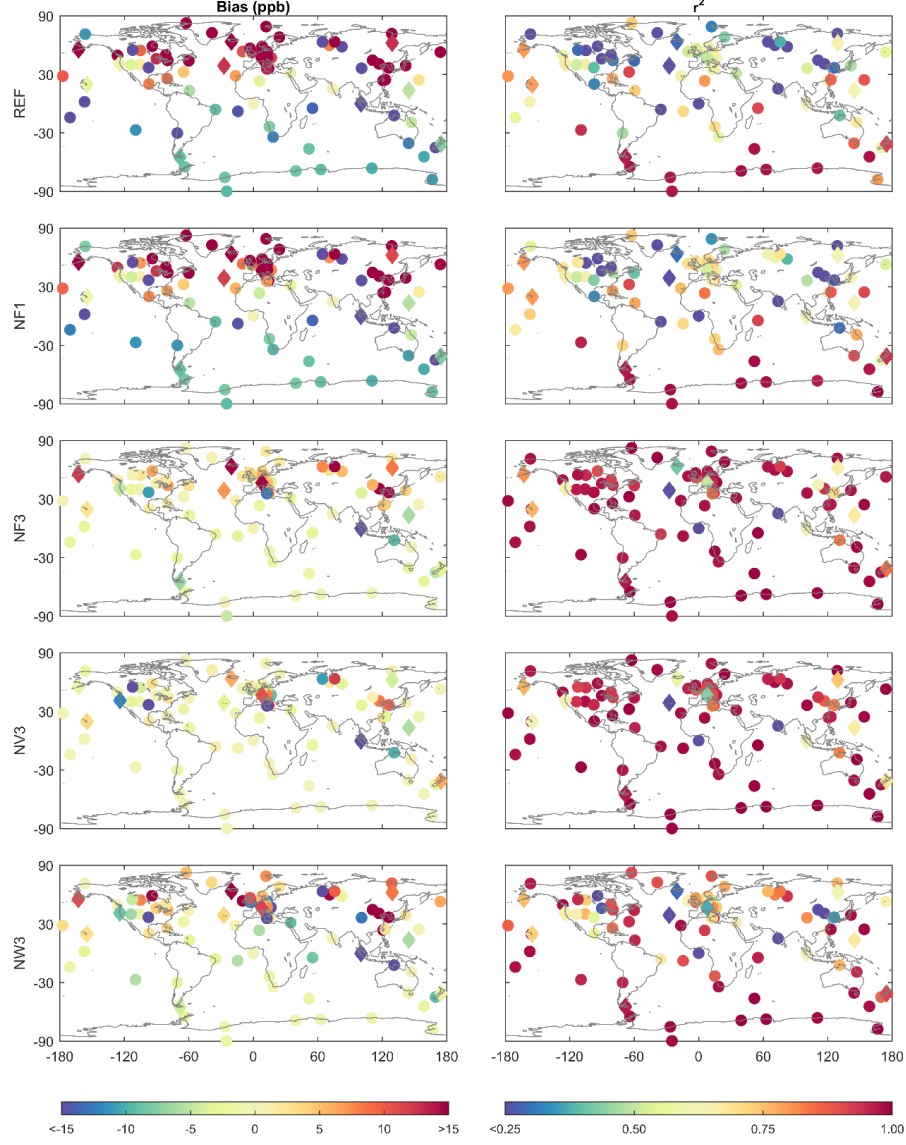

**Figure 4 Bias and coefficient of determination of modelled methane at all surface stations, including validation stations (diamonds) for a selection of simulations (see Table 1 for settings). Some bias values exceed (-)15 ppb (also see Figure 3); colour limits were chosen to show the influence of nudging at the majority of stations.**



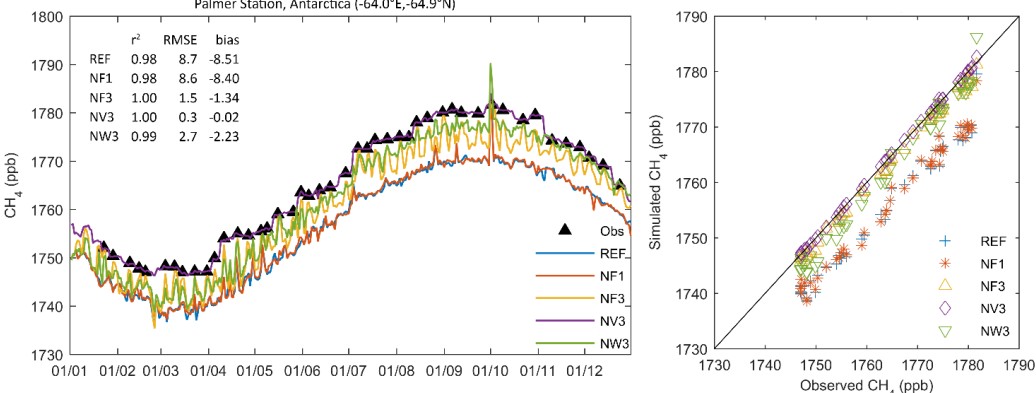

**Figure 5 Methane (ppb) as observed and modelled at Palmer Station in 2013 for a selection of FLEXPART CTM simulations, shown as time series (left) and scatter plot (right).**

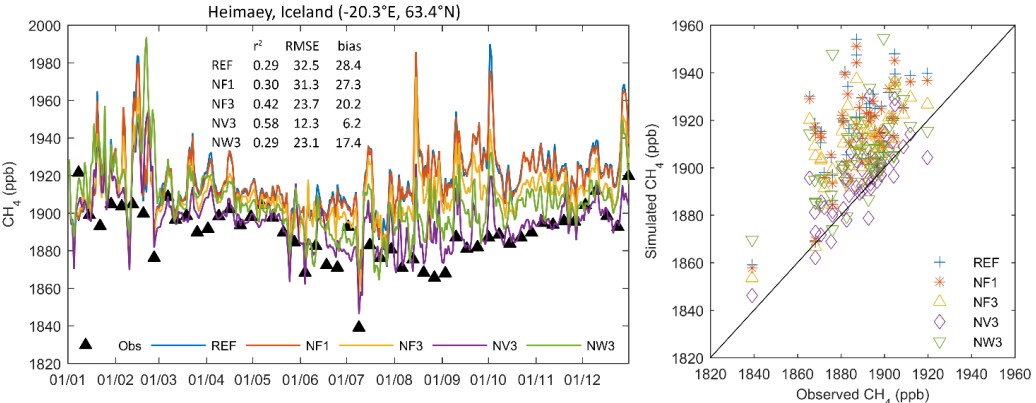

**Figure 6 Methane (ppb) as observed and modelled at Heimaey in 2013 for a selection of FLEXPART CTM simulations, shown as time series (left) and scatter plot (right).**





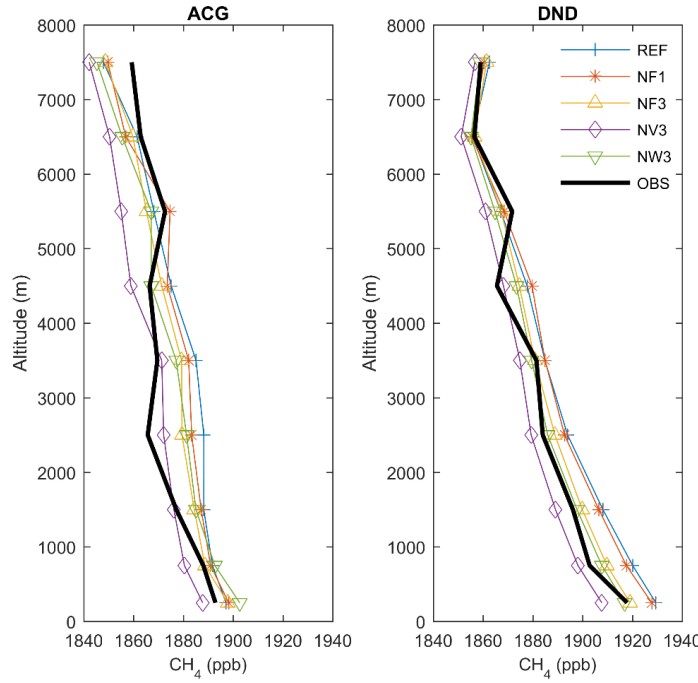


**Figure 7 Averaged annual vertical methane profiles for 2013 based on monthly averaged data at NOAA profile locations ACG (Alaska Coast Guard, United States of America) and DND (Dahlen, North Dakota, United States of America). See Figure 1 for approximate profile locations.**

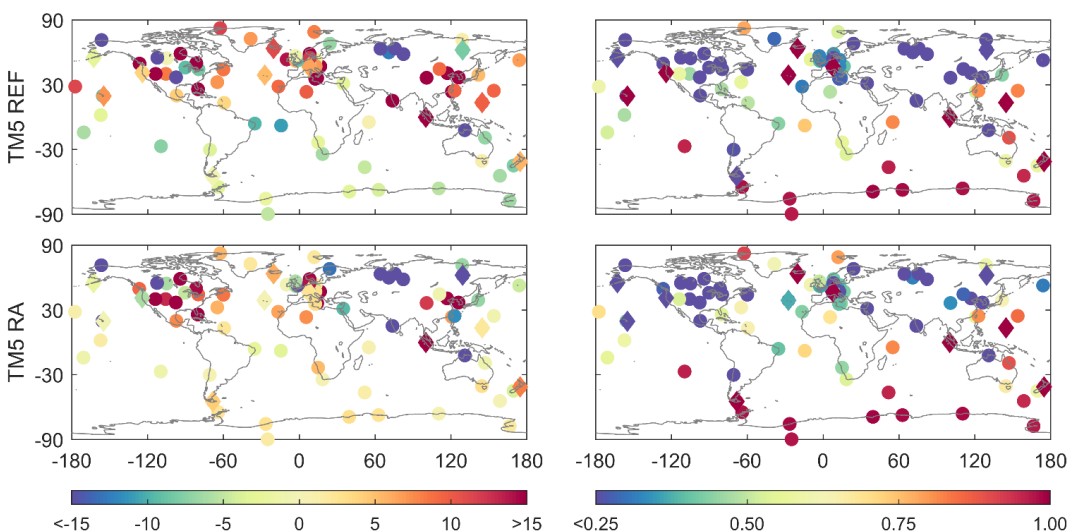

**Figure 8 Bias (ppb, left) and r² (right) of modelled methane at all surface stations for TM5 (reference, top and reanalysis, bottom) simulations.**





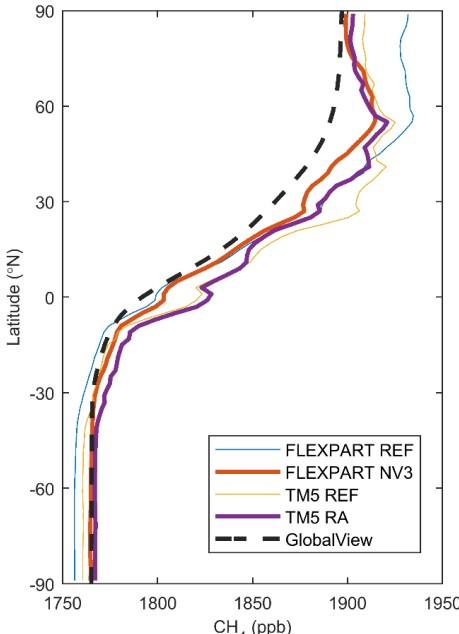

**Figure 9 Zonal mean surface methane (ppb) in 2013 based on FLEXPART CTM, TM5 and GLOBALVIEW-CH4.**





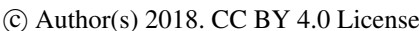

**590** **Figure 10 Averaged modelled and observed vertical profiles in 2013 at 16 NOAA locations, during the YAK-Aerosib campaign and at Surgut and Novosibirsk. FLP refers to FLEXPART CTM.**





**Figure 11 Seasonal observed and modelled methane profiles in 2013 at Estevan Point (Canada). FLP refers to FLEXPART CTM.**

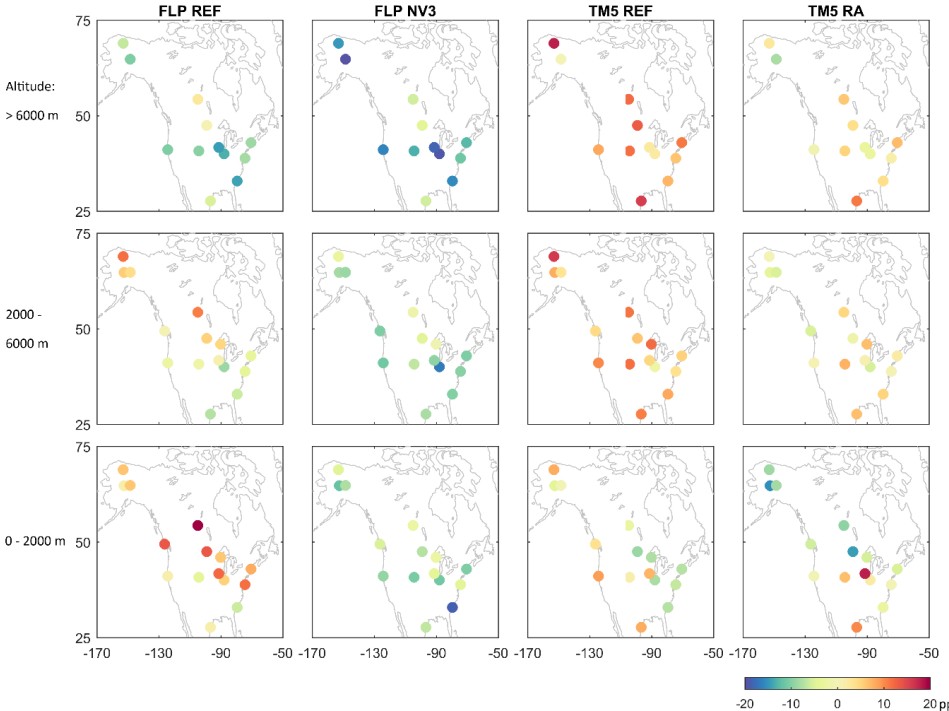

**Figure 12 Model bias of FLEXPART CTM (abbreviated FLP) and TM5 at three altitude ranges for NOAA profiles in 2013 in North America.**