# Peer review of "Three-dimensional methane distribution simulated with FLEXPART 8-CTM-1.1 constrained with observation data"

_Geoscientific Model Development, 2018_

## Referee Comment (RC1) · Anonymous Referee #1 · 20 Jul 2018

This paper is nicely written and addresses sharply the question whether local nudging of $CH_4$ mixing ratios at the surface towards stationary observations does improve the overall 3-dimensional performance.

It is an attractive method to directly include several local surface observations instead of averaged two dimensional fields. The authors provided a coherent evaluation with surface stations and aircraft profiles. However, I also would be interested in the comparison with spatially inclusive and comprehensive data sets as satellite data, which would further evaluate the whole column.

In the following I list a couple of questions concerning the manuscript.

- On page 2 line 44 you state that inverse modeling approaches need GHG concentrations as input. That is likely the case, however, I wonder if you mean the inverse modeling of $CH_4$. In this case, nudging towards observations as in your case would influence the a priori. Maybe you could be a little more specific in what kind of situation the 3D concentrations are needed. Moreover, what do you think about the use of these 3D data sets for radiation simulations?

- Introduction: To my knowledge there are a couple of models which perform nudging of GHGs. Could you list some and describe the difference or similarity to your method?

- Page 3 line 108-110: Where does those fields come from? Simulations of Chemistry-Climate Models? I understand that some reference work is not published, but the loss of methane is an important part in the simulations and needs to be replicable.

- Page 3 line 118-121: This is a very long sentence. However, the information it holds is very crucial (simulation period). Please reformulate. Furthermore, why is the scaling factor applied?

- Could you also invest a sentence in this paragraph on the introduction of your reference and sensitivity simulations? It gets lost in the results. I like the table 1 as an overview, however, it is difficult to understand without a short explanation (what is important?).

- page 8 line 307: Have you considered the methane lifetime? Compared the one of FLEXPART and TM5? What about OH and temperature?

- page 9 line 333: Are the simulated profiles sampled to the campaign profiles? Or is a certain spot chosen?

- General: If the performance of this nudging method decreases at higher altitude, I am curious to what extent does this improve the 3 dimensional field of the whole atmosphere/troposphere. Since the 3D fields are part of the motivation, could you comment on that?

Technical corrections:

- page 6 line 207: What about NW1 and NW2? I would assume that it should be (NV1-3 and NW1-3).

- Table 1: Could you highlight (additional horizontal line) the simulations with variable spatial width? What do the variable temporal width (NW1-3) mean? How are they constructed?

- Fig 11, legend: Should it be TM5 RA instead of RM5 RA?

---

## Referee Comment (RC2) · Anonymous Referee #2 · 15 Aug 2018

This is a good and valid paper without major flaws prohibiting publication. The methodology is sound. The writing is clear and well-structured. The nudging method seems to be a cost-effective and robust way to improve the simulation of 3-D field CH4 concentration. However, I found it's a bit hard to follow the Results part as some of the statement lacks explanation and conclusive sentence. I also have a few questions listed below:

- It seems the simulation of vertical profile didn't get improved after nudging. How does this affect the potential applications of 3-D CH4 concentration from FLEXPART?

- How large is the influence of priori CH4 fluxes on the model performance? It would be

helpful to address it more clearly as this will help readers from a broader background.

- How does modeled CH4 distribution compared with satellite observations like GOSAT? It would be interesting to see the evaluation against this spatially comprehensive dataset.

Specific comments:

Line 42: reference needed.

Section 2.1: more details about the setup of the methane sinks are needed.

Line 119: Please explain the reason for why applying a single global scaling factor is necessary.

Line 159: What is NOAA_2004 scale. and why NIES data is needed to be converted into NOAA-2004 scale. More statements are needed to justify this treatment.

Line 177. Does the TM5 reference simulation use same priori information as FLEX-PART? Do you think it will affect the evaluation of FLEXPART with TM5?

---

## Referee Comment (RC3) · Anonymous Referee #3 · 16 Aug 2018

The article by Groot Zwaaftink et al deals with nudging of modelled methane concentration fields towards surface observation data. It is an interesting and important contribution in its field, and suitable for publication in GMD. The paper is well written. I have a couple of minor comments, which are given in the following:

I agree that using spatially inclusive data sets, such as satellite data, would provide a valuable addition for evaluation of the model results. The data has limitations (biases etc.) but still they could possibly be used for retrieving e.g. latitudinal band averages of the column concentrations and compared to corresponding model products, to see e.g. the changes in the north-south gradient and annual cycle.

[Figure]

Moreover, I am missing an example (figure?) of how the effect of nudging is seen on the evolution of concentrations at different altitudes over a time period of days/weeks.

You mention in line 123 that you save the output in 2x2 degree resolution. Why is this resolution chosen, though you have the ability for 1x1 resolution ? Generally, how would the results change if you made the simulations in a higher spatial resolution? And the kernel settings, e.g. choices for the spatial nudging kernel sizes?

Is the vertical kernel size hz (Eq. 2) related to tropospheric boundary layer height? Could you use e.g. model predictions of boundary layer height for hz? Or add night/day variation to hz? Boundary layer height might not be meaningful for all stations, as they are located at different altitudes and sampling routines vary, but should there be some variation in the hz from station to station ?

Seems that quite much trust is given to the stations with low standard deviation, as in NV3 the concentrations are forced to follow observations at Palmer Station almost from point to point (Fig 5). The bias is corrected, but the concentration is forced to stay close to the value given by the observation, which is made only once per week. Could you elaborate this a little bit more, you say that this is a more realistic choice for a remote low emission site, but is the model ability to make predictions and fill in the gaps then lost?

---

## Author Comment (AC1) · 17 Sep 2018

**Response to reviews for "Three-dimensional methane distribution simulated with FLEXPART 8-CTM-1.1 constrained with observation data", gmd-2018-117**

We thank all referees for their useful comments and suggestions. Our response to all comments is given below.

**Anonymous Referee #1**

This paper is nicely written and addresses sharply the question whether local nudging of CH4 mixing ratios at the surface towards stationary observations does improve the overall 3-dimensional performance.
It is an attractive method to directly include several local surface observations instead of averaged two dimensional fields. The authors provided a coherent evaluation with surface stations and aircraft profiles. However, I also would be interested in the comparison with spatially inclusive and comprehensive data sets as satellite data, which would further evaluate the whole column.
In the following I list a couple of questions concerning the manuscript.

> We have considered including satellite data, but the potential biases of such data compared to observations are so large that we found it rather complicates than clarifies the evaluation and results of our model. For an impression, we show in the figure below the bias of GOSAT data compared to aircraft observations at different altitude ranges. Mean absolute bias of all profiles for these altitudes ranged from roughly 7 to 12 ppb for FLEXPART-CTM as well as GOSAT methane fields.

[Figure]

*Figure 1 Bias (ppb) of modelled fields and satellite fields of methane compared to NOAA aircraft profiles at several locations at 3 altitude ranges. The model simulations cover data for 2013. GOSAT data (Product L4B, Three-dimensional global distribution of CH4 concentration) was only available for months January until September in 2013.*

On page 2 line 44 you state that inverse modeling approaches need GHG concentrations as input. That is likely the case, however, I wonder if you mean the inverse modeling of $CH_4$. In this case, nudging towards observations as in your case would influence the a priori. Maybe you could be a little more specific in what kind of situation the 3D concentrations are needed. Moreover, what do you think about the use of these 3D data sets for radiation simulations?

> We wrote that "Inverse modelling of greenhouse gas emissions often requires global 3D greenhouse gas concentrations as input". For regional inverse modelling of greenhouse gases, 3D fields of GHG concentrations are used to account for the influence of mixing ratios outside the time and space domains of the regional model on the observations. This may be done by coupling the regional model to the 3D fields to calculate the transport of gas from outside the domain to the observation points. This is independent of the a priori, but is a source of uncertainty in the inversion. Applications of the 3D concentration fields reach further than inverse modelling of greenhouse gases (which was our main motivation). Radiation simulations is another relevant example and we added this in our introduction.

• Introduction: To my knowledge there are a couple of models which perform nudging of GHGs. Could you list some and describe the difference or similarity to your method?

> We are not aware of any 3D model that is nudged to observed GHG concentrations. There are atmospheric transport models that are nudged to observed meteorology, or nudging between model scales, and there is for example the data assimilation of $CO_2$ in one of the ECMWF models but this is different (far more computationally demanding etc.).

• Page 3 line 108-110: Where does those fields come from? Simulations of Chemistry-Climate Models? I understand that some reference work is not published, but the loss of methane is an important part in the simulations and needs to be replicable.

> The monthly OH concentration fields are from the GEOS-Chem model (Bey et al., 2001). Note that any errors in the estimated loss due to e.g. OH are corrected for by nudging to observed CH4 mixing ratios. We added this information in the manuscript.

• Page 3 line 118-121: This is a very long sentence. However, the information it holds is very crucial (simulation period). Please reformulate. Furthermore, why is the scaling factor applied?

> The scaling factor was applied to remove the bias from the initial fields used in our simulations. We reformulate to:
>
> "A spin-up simulation without nudging was run for years 2000-2012. At the end of this spin-up a single global scaling factor was applied to the simulated methane molar mixing ratios derived by a comparison to surface observations for the year 2012. This scaling allowed us to remove part of the bias from the spin-up. The sensitivity analysis and evaluation of FLEXPART CTM and the nudging method was made for 2013."

Could you also invest a sentence in this paragraph on the introduction of your reference and sensitivity simulations? It gets lost in the results. I like the table 1 as an overview, however, it is difficult to understand without a short explanation (what is important?).

We added a brief description here. An introduction to table 1 is given in section 2.2 after explaining the nudging routine.

page 8 line 307: Have you considered the methane lifetime? Compared the one of FLEXPART and TM5? What about OH and temperature?

We have not looked further into differences in TM5 and FLEXPART-CTM. There are indeed many components that can be relevant here. However, we do not wish to make this into a study comparing FLEXPART CTM and TM5 in detail, but like to focus on the nudging routine and final background fields. We added other factors that may be important.

page 9 line 333: Are the simulated profiles sampled to the campaign profiles? Or is a certain spot chosen?

Yes, simulated profiles are sampled at campaign observations. We now repeat this information in the results section.

General: If the performance of this nudging method decreases at higher altitude, I am curious to what extent does this improve the 3 dimensional field of the whole atmosphere/troposphere. Since the 3D fields are part of the motivation, could you comment on that?

Indeed we find most improvement of modelled fields near the surface, up to heights of 6 km, and less confidence should be given to the fields at higher altitudes. However, we also show that at higher altitudes model performance is similar to TM5. Although there are larger uncertainties for these regions, one should consider that there are currently not many alternatives.

**Technical corrections:**
page 6 line 207: What about NW1 and NW2? I would assume that it should be (NV1-3 and NW1-3).

NW1 should be include here. Corrected.

Table 1: Could you highlight (additional horizontal line) the simulations with variable spatial width? What do the variable temporal width (NW1-3) mean? How are they constructed?

We added additional horizontal lines. The variable temporal width is explained in section 2.2.

Fig 11, legend: Should it be TM5 RA instead of RM5 RA?

Yes. Corrected

**Anonymous Referee #2**

This is a good and valid paper without major flaws prohibiting publication. The methodology is sound. The writing is clear and well-structured. The nudging method seems to be a cost-effective and robust way to improve the simulation of 3-D field CH4 concentration. However, I found it's a bit hard to follow the Results part as some of the statement lacks explanation and conclusive sentence.

I also have a few questions listed below:

- It seems the simulation of vertical profile didn't get improved after nudging. How does this affect the potential applications of 3-D CH4 concentration from FLEXPART?

Improvements are mainly seen below 6 km altitude, which is clearly stated in the manuscript. Of course this should be considered when using these fields for applications and other alternatives might be more appropriate if the interest is only in the upper troposphere. However, for applications where the lower atmosphere is of interest, we show that surface results are better than optimized TM5 fields and performance based on vertical profiles is similar. We therefore see an advantage in the nudging method anyway. Especially for our purpose of regional inverse modelling based on surface observations, where it is crucial to have limited bias in the background fields in this region and the bias at higher altitudes is similar to other alternatives currently available.

- How large is the influence of priori CH4 fluxes on the model performance? It would be helpful to address it more clearly as this will help readers from a broader background.

The a priori CH4 fluxes are important for model performance. The nudging is only able to correct the modelled values to some extent and does not make accurate a priori fluxes redundant. However, note that the performance of FLEXPART-CTM with default a priori fluxes and nudging surpasses that of the TM5 with optimized fluxes, demonstrating that although the a priori fluxes have some influence, the nudging method is still valid for deriving accurate 3D concentration fields.

- How does modeled CH4 distribution compared with satellite observations like GOSAT? It would be interesting to see the evaluation against this spatially comprehensive dataset.

We compared the NOAA aircraft profiles of methane to GOSAT data, similar to our model comparison. Figure 1 (response review 1) shows that the bias of GOSAT data compared to the aircraft observations generally exceeds the bias of the different model simulations. We therefore have little confidence in an evaluation against this dataset. Although we agree that a spatial dataset is interesting to include in our evaluation, we find it doubtful that we can retrieve clear conclusions from this comparison and think it will overly complicate the manuscript.

Specific comments:

Line 42: reference needed.

References are given for specific examples following line 42.

Section 2.1: more details about the setup of the methane sinks are needed.

We added a reference for the OH fields.

Line 119: Please explain the reason for why applying a single global scaling factor is necessary.

This was needed because we found a bias of the fields after the spin-up period (without nudging). With a single global scaling factor we could remove large parts of this bias and start with improved initial conditions for our simulation. We added a comment on this.

Line 159: What is NOAA_2004 scale. and why NIES data is needed to be converted into NOAA-2004 scale. More statements are needed to justify this treatment.

This is a calibration scale for methane and is used to make observations intercomparable. We added a reference for clarification. All data we used for the 2013 simulations has been converted to this scale. The conversion of NIES data has been mentioned explicitly because different conversion methods exist.

Line 177. Does the TM5 reference simulation use same priori information as FLEXPART? Do you think it will affect the evaluation of FLEXPART with TM5?

No, there are some differences in the a priori information. This was mentioned as a reason for differences in line 307. We now repeat this in the conclusions. Because of this we can only evaluate the final methane fields compared to observations, considering it a combination of a priori information and optimized transport modelling. Only looking at the improvements in each of the models would be somewhat misleading. That is, the improvements, a nudging or other optimization routine needs to add, will partly depend on a priori information and a larger improvement of either of the models compared to its reference could be due to the a priori information rather than the effectiveness of the optimization scheme. Simulations with the same a priori information could exclude this effect, as suggested in the conclusions. A priori information used for both model simulations is quite similar in our study.

**Anonymous Referee #3**

The article by Groot Zwaaftink et al deals with nudging of modelled methane concentration fields towards surface observation data. It is an interesting and important contribution in its field, and suitable for publication in GMD. The paper is well written. I have a couple of minor comments, which are given in the following:

I agree that using spatially inclusive data sets, such as satellite data, would provide a valuable addition for evaluation of the model results. The data has limitations (biases etc.) but still they could possibly be used for retrieving e.g. latitudinal band averages of the column concentrations and compared to corresponding model products, to see e.g. the changes in the north-south gradient and annual cycle.

> Indeed we tried to include methane fields from satellite data (GOSAT). However, we found that bias of this data compared to in situ observations is so large that it is not possible to draw conclusions on accuracy of our model simulations. In Figure 1 (response review 1) we illustrate this with an example of bias of all models and GOSAT data compared to aircraft observations at 3 altitude ranges.

Moreover, I am missing an example (figure?) of how the effect of nudging is seen on the evolution of concentrations at different altitudes over a time period of days/weeks.

> We now included Figure 3 in the manuscript, showing how nudging influences methane concentrations over 31 days at altitudes up to 16 km.

You mention in line 123 that you save the output in 2x2 degree resolution. Why is this resolution chosen, though you have the ability for 1x1 resolution ? Generally, how would the results change if you made the simulations in a higher spatial resolution? And the kernel settings, e.g. choices for the spatial nudging kernel sizes?

> Higher resolution could improve results in regions with highly variable methane concentrations and especially the comparison to point observations. However, the 2x2 degree resolution already includes the level of detail necessary for our applications and is a compromise to limit computational efforts and data storage.
> The nudging kernel sizes are independent of the output resolution because we use a Lagrangian model.

Is the vertical kernel size hz (Eq. 2) related to tropospheric boundary layer height? Could you use e.g. model predictions of boundary layer height for hz? Or add night/day variation to hz? Boundary layer height might not be meaningful for all stations, as they are located at different altitudes and sampling routines vary, but should there be some variation in the hz from station to station?

> The vertical kernel size is the same for all locations in NV3 (300 m). Thus, for daytime observations the kernel most likely does not exceed the boundary layer height for the majority of observation locations. Including the boundary layer height and night/day variation is technically possible but we did not do this because it may overly complicate the nudging routine while the sensitivity analysis showed that the horizontal extent of the kernel is more important for our model results.

Seems that quite much trust is given to the stations with low standard deviation, as in NV3 the concentrations are forced to follow observations at Palmer Station almost from point to point (Fig 5). The bias is corrected, but the concentration is forced to stay close to the value given by the observation, which is made only once per week. Could you elaborate this a little

bit more, you say that this is a more realistic choice for a remote low emission site, but is the model ability to make predictions and fill in the gaps then lost?

Palmer station is an extreme example of the nudging routine. If the temporal kernel width exceeds the gaps between observations, the model abilities are indeed limited. However, strong deviations between observations and model will not always be compensated by the nudging routine and the model does not necessarily follow the observations from point to point. The example below (Figure 2) shows a different site, where model values adjust more gradually.

[Figure]

*Figure 2 Observed and FLEXPART CTM simulated methane values at Mahe Island (Seychelles) without nudging (REF) and with nudging (NV3).*